Estimating risks of importation and local transmission of Zika virus infection

Nah Kyeongah 1 2 3
Mizumoto Kenji 1 3 4
Miyamatsu Yuichiro 1 2 3
Yasuda Yohei 1
Kinoshita Ryo 1 2 3
Nishiura Hiroshi 1 2 3 nishiurah@gmail.com
1 Graduate School of Medicine, The University of Tokyo , Tokyo , Japan
2 CREST, Japan Science and Technology Agency , Saitama , Japan
3 Graduate School of Medicine, Hokkaido University , Sapporo , Japan
4 Graduate School of Arts and Sciences, The University of Tokyo , Tokyo , Japan
Benelli Giovanni
Electronic publication date: 2016 Apr 5
Publication date: 2016
Volume: 4
Electronic Location ID: e1904
Received 2016 Feb 11; Accepted 2016 Mar 16
Copyright: ©2016 Nah et al.
Copyright year: 2016
Copyright holder: Nah et al.
License: This is an open access article distributed under the terms of the Creative Commons Attribution License, which permits unrestricted use, distribution, reproduction and adaptation in any medium and for any purpose provided that it is properly attributed. For attribution, the original author(s), title, publication source (PeerJ) and either DOI or URL of the article must be cited.
License URL: https://creativecommons.org/licenses/by/4.0/

Keywords: Zika virus, Importation, Risk, Mathematical model, Transmission, Statistical estimation, Network, Epidemiology

Funding: Japanese Society for the Promotion of Science 26670308 26700028 Japan Agency for Medical Research and Development Japan Science and Technology Agency CREST program RISTEX program for Science of Science, Technology and Innovation Policy Japanese Society for the Promotion of Science 15K20936 HN received funding support from the Japanese Society for the Promotion of Science (JSPS) KAKENHI Grant Numbers 26670308 and 26700028, Japan Agency for Medical Research and Development, the Japan Science and Technology Agency (JST) CREST program and RISTEX program for Science of Science, Technology and Innovation Policy. KM received funding support from the Japanese Society for the Promotion of Science (JSPS) KAKENHI Grant Number 15K20936. The funders had no role in study design, data collection and analysis, decision to publish, or preparation of the manuscript.

==============================
Background. An international spread of Zika virus (ZIKV) infection has attracted global attention. ZIKV is conveyed by a mosquito vector, Aedes species, which also acts as the vector species of dengue and chikungunya viruses.

Methods. Arrival time of ZIKV importation (i.e., the time at which the first imported case was diagnosed) in each imported country was collected from publicly available data sources. Employing a survival analysis model in which the hazard is an inverse function of the effective distance as informed by the airline transportation network data, and using dengue and chikungunya virus transmission data, risks of importation and local transmission were estimated.

Results. A total of 78 countries with imported case(s) have been identified, with the arrival time ranging from 1 to 44 weeks since the first ZIKV was identified in Brazil, 2015. Whereas the risk of importation was well explained by the airline transportation network data, the risk of local transmission appeared to be best captured by additionally accounting for the presence of dengue and chikungunya viruses.

Discussion. The risk of importation may be high given continued global travel of mildly infected travelers but, considering that the public health concerns over ZIKV infection stems from microcephaly, it is more important to focus on the risk of local and widespread transmission that could involve pregnant women. The predicted risk of local transmission was frequently seen in tropical and subtropical countries with dengue or chikungunya epidemic experience.

Introduction

Zika virus belongs to the virus family Flaviviridae and genus Flavivirus, initially isolated from Zika forest in Uganda (Dick, 1952). Related to other arboviruses such as chikungunya, dengue, yellow fever, Japanese encephalitis and West Nile viruses, the virus had circulated in sub-Saharan African, South and Southeast Asian and South Pacific countries. Once reaching South America, an international spread of Zika virus (ZIKV) infection has attracted global attention, as a large epidemic of ZIKV infection in Brazil was considered to have been associated with a remarkable increase in the number of microcephaly cases that are suspected to have been caused by ZIKV infection among pregnant women (Staples et al., 2016). The causal relationship between ZIKV infection and microcephaly has yet to be fully established, but an increasing amount of scientific evidence strongly supports the epidemiological link of ZIKV with microcephaly (Mlakar et al., in press; Brasil et al., in press; Nishiura et al., 2016b). Considering the expected impact on neurological manifestations including microcephaly and Guillain-Barré Syndrome (GBS), the World Health Organization issued the Public Health Emergency of International Concern (PHEIC) on 1 February 2016.

ZIKV shares self-limiting clinical signs and symptoms with dengue virus (DENV) and chikungunya virus (CHIKV) (Ioos et al., 2014). Moreover, these viruses are also conveyed by the common mosquito vector, Aedes species. Technically, the infected cases could be observed in any part of the world in the presence of Aedes species, but the risk of transmission in Latin American and Caribbean countries has been anticipated to be high (Musso, Cao-Lormeau & Gubler, 2015; Bogoch et al., 2016). The transmission potential of ZIKV infection has been shown to be comparable to those of DENV and CHIKV (Nishiura et al., 2016a). Considering that there are no specific treatments for these arboviruses, it is of utmost importance to intervene transmission by preventing mosquito biting (e.g., the use of repellent) or controlling Aedes species, most notably by targeting highly invasive species including Aedes aegypti and Aedes albopictus. The control option of Aedes spp. includes source reduction (i.e., removal of water and covering water storage containers), the use of ovitrap, chemical control, bioinsecticides and biological control (e.g., mosquitofish) (Anonymous, 2016), while recent approaches include transgenic and symbiont-based approaches and the use of plant-borne molecules (Adelman & Tu, 2016; Benelli, 2016; Benelli & Mehlhorn, 2016).

To plan for forthcoming potential local transmission at each country level, it is fruitful to understand the actual risk of Zika virus epidemic in a quantitative manner. The present real-time study aimed to identify countries at high risk of importing ZIKV infection and also of local transmission as a part of risk assessment practice and for improved understanding of the predicted risk of the spread, with distinct differentiation of ZIKV importation alone from that of local transmission.

Materials & Methods

Epidemiological data

Arrival times of ZIKV importation, i.e., the time at which first ZIKV case was imported, in each imported country was collected from publicly available data sources (see Supplemental Information). The criterion to include a country as imported was the presence of a report based on confirmatory diagnosis made either by serological testing or detection of virus RNA. When ZIKV was isolated from mosquitoes or primates other than humans, it was considered that the country had already imported the virus, because the virus isolation implies that the local transmission had taken place. For each imported country, the arrival time was defined as the week in which the importation event had happened. When monthly or yearly data were only available or when a seroepidemiological study identified the local transmission, the median week was used as the week of importation. The latest time at which the importation week was systematically examined was 31 January 2016.

In addition to the importation data, we collected the list of countries with local transmission of ZIKV from the European Centre for Disease Prevention and Control (2016) and Gatherer & Kohl (2016). ECDC resource has specified countries or territories with reported confirmed autochthonous cases of ZIKV infection in the past 9 months. Since the European Centre for Disease Prevention and Control (2016) list misses countries that experienced local transmission in 2014 or earlier, we extracted the list of those earlier countries from Gatherer & Kohl (2016). Again, the virus isolation from mosquitoes or primates other than humans was regarded as a signature of local transmission.

As additional input of prediction models, an open source airline transportation network data were extracted. Using the Global Flights Network (2016) derived from the OpenFlights database as of 10 November 2014 (Contentshare, 2016), we obtained the total number of flight routes between each pair of countries (a total of 230 airports and 4,600 flight routes). Furthermore, the country-specific presence of Aedes species, especially focusing on Aedes albopictus, Aedes aegypti and Aedes africana, was explored (Kraemer et al., 2015; The Walter Reed Biosystematics Unit, USA, 2016). The list of countries with at least one of the abovementioned Aedes species was prepared, because the local human-vector-human transmission of ZIKV infection occurs only where the vector species are present. Similarly, we also retrieved the country-specific data regarding DENV and CHIKV epidemic (Centers for Disease Control and Prevention, 2016a; Centers for Disease Control and Prevention, 2016b), because they share major vector species with ZIKV and their epidemics indicate that the vector species have been actively involved in local transmission. Moreover, it has been explicitly discussed that international spread of Zika virus followed the geographic path of CHIKV (Musso, Cao-Lormeau & Gubler, 2015). The presence of Aedes species, DENV and CHIKV were all dealt with as dichotomous variable.

Prediction model

We estimated the risk of importing ZIKV and that of local transmission using a survival analysis model. Let T be a continuous random variable with probability density function of the time from importation in Brazil to importation in country j, fj(t), and cumulative distribution function Fj(t) = Pr(T < t). The function Fj(t) describes the probability that ZIKV has been already imported to a country j by time t. The time t = 0 corresponds to the time at which ZIKV infection was first recognized in Brazil and started to rapidly spread across countries (i.e., week 12 of 2015). We parameterize the hazard function λ(t) using the effective distance, the metric derived from the airline transportation network (Contentshare, 2016). Let {n1, n2, …nl} be the sequence of transit countries at which a traveler starting from country n1 (with a destination nl) stops over. The length of path of that travel is l. The effective length of the path, dn1nl, is defined by (1) dn1nl=l− log∏k=1l−1Pnk+1nk,

where Pji denotes the conditional probability that an individual that left i moves to j. Assuming that the number of passengers is identical among all international flights, the transition matrix is calculated as Pji=wji∑kwki, where wki is the number of direct flights from country i to country k per unit time derived from airline transportation network data (Contentshare, 2016). Finally, the effective distance mj of a country j from the country of origin (i.e., Brazil) is calculated as the minimum of the all possible effective lengths of path that goes from the origin country to the country j. The effective distance has been known as an excellent predictor of the arrival time of SARS (severe acute respiratory syndrome) and influenza pH1N1 2009 (Brockmann & Helbing, 2013). Considering that the effective distance is a critical indicator of the risk of importation, we assume that the hazard function is an inverse of the effective distance, i.e., (2) λjt=kmj,

where k is a constant. Using the hazard function, the abovementioned density function is modeled as (3) fjt=λjtexp−∫0tλjsds.

Modeling fj(t) in this way, the mean arrival time of the ZIKV to a country j would be proportional to the effective distance from the origin of spread (i.e., Brazil in our case study).

Using the calculated risk of importation, we subsequently model the country specific risk of local transmission. Let pj be the conditional probability of ZIKV transmission given an importation event in a country j. Since our time scale of examining the risk of importation is much longer than the generation time of ZIKV infection, we ignore the time-lag that is required for observing the transmission cycle, and the risk of the local transmission of ZIKV in a country j is modeled as (4) gjt=pjfjt.

In order to estimate pj, the conditional probability of experiencing local transmission, we use datasets of the presence of Aedes species, CHIKV and DENV in each country. More specifically, we model pj by employing a logit model (5) pj=11+ exp−a0+ ∑iβixji,

where a0 is an intercept, xji is the dichotomous variable that describes the presence of Aedes mosquitoes (for i = 1), CHIKV (for i = 2) and DENV (for i = 3) in country j, and βi represents the coefficient for corresponding variable i.

Parameter estimation and risk assessment

We estimate parameters k, a0 and three βi using a maximum likelihood method. Since we start the clock from week 12, 2015 originated from Brazil (Campos, Bandeira & Sardi, 2015), countries that had imported ZIKV in advance (e.g., Africa and South Pacific) were removed before the implementation. In total, we predict the risks of importation and local transmission among a total of 189 countries. The model was fitted at week 46 since importation in Brazil (and the week 46 corresponds to the latest time at which our systematic survey of importation dates was completed in January 2016). Although five countries (Aruba, Trinidad and Tobago, Marshall Islands, Saint Vincent and the Grenadines, and American Samoa) were specified as countries with local transmission by ECDC data source (European Centre for Disease Prevention and Control , 2016), these countries have experienced importation and confirmation in February 2016, and were thus removed from the list of countries with importation as well as local transmission.

The likelihood of importation risk adhered to survival model. That is, among countries that have already imported ZIKV by then, the arrival time tj was used for parameter estimation. Countries that have not imported ZIKV by week 46 were dealt with as the censored observation. In other words, the survival time for those countries is considered to be at least as long as the duration of our study. The likelihood function reads (6) Lk,a0,βi;tj,xj,tm= ∏j∈Apjfjtj∏j∈B1−pjfjtj ∏j∈C1−Fjtm,

where A represents the set of countries which experienced local transmission of ZIKV by time of observation tm (31 January 2016), B represents countries which imported ZIKV before tm but without local transmission, and C is the set of countries which has not imported by tm. Once the parameter k is estimated, we are able to predict the probability of country j to experience local transmission of ZIKV by time t as (7) Gjt=pj1−e−ktmj.

We assessed the diagnostic performance of our models in predicting risks of importation and local transmission by employing the receiver operating characteristic (ROC) curve and measuring the area under the curve (AUC) (Greiner, Pfeiffer & Smith, 2000). The predictors of local transmission in Eq. (5) were varied by using 1–3 variables (by examining all possible combinations of Aedes, DENV and CHIKV), and we compared AUC to identify the best predictive model. To avoid serious collinearity among Aedes, DENV and CHIKV, Cohen’s kappa, an agreement statistic, was computed to examine correlations between two dichotomous explanatory variables. For each model, the optimal cut-off value of estimated risk was calculated using Youden’s index. While model fit was assessed in week 46, the prediction has been made in week 92 that corresponds to the end of 2016.

Table 1 The list of countries that experienced importation of Zika virus infection.

The earliest date at which an infected individual has likely to have entered the country is shown as the week counting from week 26, 1946. Extracting the dataset from the World Health Organization source, the week number of the report has been originally written and we used it as the week of importation. When the exact week was unavailable, the mid-point of the available time window was used as the week of importation. The first day of the year was set to be the first Sunday of January. One year was calculated to be exactly equal to 52 weeks in our analysis. Although three confirmed cases of ZIKV infection were reported in Chile on 2 February 2016, that data was ignored in the present study because the latest time was set at fourth week of 2016. We also did not count a case report from Easter Island, a Chilean island in the southeastern Pacific Ocean, because the island is geographically distant from other parts of Chile and has only one airport (World Health Organization, Western Pacific Region, 2016). In Australia, although one imported case is reported, there were no available sources of information to identify the date of confirmation. Instead, given that the corresponding article was received on 16 January 2013, we consider that the importation event took place in the previous year, 2012, and took the mid-point of the year as the week of importation (Kwong, Druce & Leder, 2013). In Zambia, a cross sectional study was conducted, but there was no available information to identify the date of importation or survey. Thus, the year of the acceptance of the article, 2015, was assumed as the year of importation (Babaniyi et al., 2015).

Country	Year	Week	Weeks since Uganda	Country	Year	Week	Weeks since Uganda	
Countries up to Brazil				Countries after Brazil			
Uganda	1947	26	0	Vanuatu	2015	17	3,527	
Tanzania	1948	26	52	Sweden	2015	28	3,538	
Indonesia	1951	26	208	Fiji	2015	33	3,543	
Malaysia	1951	26	208	Samoa	2015	37	3,547	
India	1952	26	260	Colombia	2015	42	3,552	
Philippines	1953	26	312	Suriname	2015	45	3,555	
Egypt	1953	26	312	El Salvador	2015	47	3,557	
Thailand	1954	26	364	Guatemala	2015	47	3,557	
Vietnam	1954	26	364	Mexico	2015	48	3,558	
Angola	1960	22	672	Venezuela	2015	48	3,558	
Kenya	1967	26	1,040	Netherlands	2015	48	3,558	
Ethiopia	1967	26	1,040	Panama	2015	48	3,558	
Somalia	1967	26	1,040	Paraguay	2015	48	3,558	
Gabon	1967	26	1,040	Honduras	2015	51	3,561	
Nigeria	1968	26	1,092	Cape Verde	2015	51	3,561	
Central African Republic	1968	26	1,092	Spain	2015	52	3,562	
Senegal	1968	26	1,092	Puerto Rico	2016	1	3,563	
Sierra Leone	1972	26	1,300	Martinique	2016	2	3,564	
Pakistan	1980	26	1,716	French Guiana	2016	2	3,564	
Cote d’Ivoire	1980	26	1,716	Finland	2016	2	3,564	
Burkina Faso	1981	26	1,768	United Kingdom	2016	3	3,565	
Micronesia	2007	26	3,120	Taiwan	2016	3	3,565	
United States	2008	26	3,172	Bolivia	2016	3	3,565	
Cameroon	2010	26	3,276	Ecuador	2016	3	3,565	
Cambodia	2010	34	3,284	Haiti	2016	3	3,565	
Australia	2012	26	3,380	Guadeloupe	2016	3	3,565	
Canada	2013	5	3,411	Barbados	2016	3	3,565	
Germany	2013	26	3,432	Guyana	2016	3	3,565	
French Polynesia	2013	46	3,452	Argentina	2016	4	3,566	
Japan	2013	50	3,456	Peru	2016	4	3,566	
Norway	2013	50	3,456	Portugal	2016	4	3,566	
Italy	2014	1	3,459	Austria	2016	4	3,566	
New Caledonia	2014	1	3,459	Costa Rica	2016	4	3,566	
Cook Islands	2014	9	3,467	Switzerland	2016	4	3,566	
Solomon Islands	2014	12	3,470	Dominican Republic	2016	4	3,566	
Zambia	2014	26	3,484	Jamaica	2016	4	3,566	
Belgium	2014	38	3,496	Denmark	2016	4	3,566	
Brazil	2015	12	3,522	Virgin Islands	2016	4	3,566	
				Nicaragua	2016	4	3,566	
				Tonga	2016	4	3,566	

Results

Table 1 shows the list of countries that have experienced ZIKV infection along with arrival time. A total of 78 countries with imported case(s) have been identified, with the arrival time ranging from 52 to 3566 weeks since the first ZIKV was identified in Uganda, 1947. After the introduction to Brazil in early 2015, 39 countries have experienced importation by the end of January 2016. Originally, we had defined t = 0 as the week of illness onset of first identified ZIKV case in 1947, Uganda (Dick, 1952), but the country was not specified as the origin, because a long time has passed since the emergence and the path of global spread must not have followed a static network. Note that 39 countries which had already imported ZIKV prior to importation event in Brazil were excluded from the analysis.

Using Brazil, 2015 as the origin (Figs. 1A and 1C), the model appeared to crudely capture the risk of importation (AUC = 0.84 (95% CI [0.69–1.00])). Maximum likelihood estimate of the constant parameter k was 0.044 (95% CI [0.031–0.059]). Sensitivity and specificity were estimated at 77.5% (95% CI [64.6–90.4]) and 85.9% (95% CI [80.3–91.5]), respectively. Figure 2 shows the global distribution of the risk of importation. High risks of importation are identified in South America and the western part of European countries. Among the top 30 countries predicted at high risk of ZIKV importation in the end of 2016 (Fig. 3), 18 countries (60%) had already imported ZIKV before week 46.

Figure 1 Predicted risks of ZIKV infection.

(A–B) Distribution of estimated risks of importation and local transmission by country. The effective distance was used for (A), while the presence of dengue and chikungunya viruses was additionally considered for (B). (C–D) Receiver operator characteristic curves of predicted risks of importation and local transmission. (C) shows the evaluation results of the risk of importation that rested on the effective distance from Brazil, while (D) shows the risk of local transmission additionally accounted for dengue and chikungunya virus epidemic data.

Figure 2 Global distribution of risks of importation and local transmission with Zika virus.

(A) The importation risk of ZIKV by week 92 is colored by intensity (0–20%, 20–40%, 40–60%, 60–80%). The origin country, Brazil, is colored in grey. Other additional countries colored in grey were excluded, because they experienced importation of ZIKV infection prior to the event in Brazil (week 12, 2015). (B) The local transmission risk of ZIKV infection by week 92 accounting for dengue and chikungunya epidemic data. The local transmission risk of ZIKV is colored by intensity (0–15%, 15–30%, 30–45%, 45–60%). The origin country, Brazil, is colored in grey. Other additional countries colored in grey were excluded, because they experienced importation of ZIKV infection prior to the event in Brazil (week 12, 2015).

Figure 3 Countries at high risk of ZIKV infection.

(A–B) List of top 30 countries with the estimated highest risks. (A) shows the risk of importation, while (B) shows the risk of local transmission. The risks shown on horizontal axes represent our estimates by the end of 2016 (week 92). Bars filled with grey represent countries that have already experienced importation of ZIKV infected case(s) by 31 January 2016 (week 46).

Table 2 shows the fitting results of various models to predict the risk of local transmission. Comparing predictive performance by AUC among models with all possible combinations of three explanatory variables, the model with CHIKV and DENV appeared to yield the greatest value of AUC (0.90 (95% CI [0.60–1.00])). Figures 1B and 1D show the distribution of the estimated risk and ROC curve, respectively. Using the best model, sensitivity was as high as 96.4% (95% CI [89.6–100.0]), while the specificity was estimated at 67.7% (95% CI [60.5–74.9]). The second best model was the one with CHIKV only.

Table 2 Predictive performance of risk models of importation and local transmission.

ID	Predicted risk (variables)	AUCa(95% CIb)	Cut-off (%)	Sensitivity (95% CIb)	Specificity (95% CIb)	
NA	Importation	0.84 (0.69,1.00)	20.6	77.5 (64.6, 90.4)	85.9 (80.3, 91.5)	
1	Local transmission (Aedes)	0.80 (0.55, 1.00)	16.2	64.3 (46.5, 82.0)	85.1 (79.6, 90.6)	
2	Local transmission (Chikungunya)	0.89 (0.62, 1.00)	18.6	71.4 (54.7, 88.2)	93.2 (89.3, 97.1)	
3	Local transmission (Dengue)	0.84 (0.54, 1.00)	17.4	67.9 (50.6, 85.2)	91.9 (87.7, 96.1)	
4	Local transmission (Aedes & Chikungunya)	0.89 (0.61, 1.00)	14.7	85.7 (72.8, 98.7)	80.1 (74.0, 86.3)	
5	Local transmission (Aedes & Dengue)	0.86 (0.55, 1.00)	13.7	82.1 (68.0, 96.3)	74.5 (67.8, 81.3)	
6	Local transmission (Chikungunya & Dengue)	0.90 (0.60, 1.00)	9.71	96.4 (89.6, 100.0)	67.7 (60.5, 74.9)	
7	Local transmission (Aedes & Chikungunya & Dengue)	0.76 (0.38, 1.00)	51.2	89.3 (77.8, 100.0)	70.8 (63.8, 77.8)	
Notes.

a AUC, area under the curve. The confidence intervals were calculated using Mann–Whitney method (Gengsheng & Hotilovac, 2008).

b CI, confidence interval.

Table 3 shows parameter estimates of the best fitted model. Compared with the absence of Chikungunya virus, the risk of local transmission in the presence of this virus was shown to be indicative of 22.9 times (95% CI [3.3–238.3]) higher. The presence of dengue was not statistically significant, but the adjusted odds ratio was estimated to be 7.7 (95% CI [1.0–73.6]). Table 4 shows correlations between two dichotomous variables as measured by Cohen’s kappa. No particular correlation that could lead to multi-collinearity (e.g., kappa > 0.60) was identified.

Table 3 Estimated parameters for describing the local transmission risk of Zika virus infection.

	Estimated values (95% CI)	Adjusted odds ratio (95% CI)	
Intercept	2.06 (−4.23, −0.51)	NA	
Chikungunya virus	3.13 (1.19, 5.47)	22.90 (3.30, 238.34)	
Dengue virus	2.04 (−0.02, 4.30)	7.68 (0.98, 73.64)	
Notes.

NA, not applicable. Results from the best model that included Chikungunya and Dengue as explanatory variables are shown. Dependent nominal variable = local transmission. CI, confidence interval.

Table 4 Agreement statistic kappa between two dichotomous variables.

Combination of variables	Cohen’s kappa, an agreement statistic (95% CIa)	
Aedes and Chikungunya	0.442 (0.327, 0.556)	
Aedes and Dengue	0.366 (0.235, 0.496)	
Chikungunya and Dengue	0.523 (0.405, 0.642)	
Notes.

a CI, confidence intervals.

Figure 2B shows the country-specific global distribution of local transmission using the best model. The high risk of local transmission is seen mainly among countries in tropical and subtropical areas. Figure 3B show the top 30 countries with high risk of local transmission by week 92, using Brazil as the origin of spread. Among the total, 19 countries (63.3%) were predicted to have already allowed local transmission before week 46.

Discussion

The present study estimated country-specific risk of importation and local transmission of ZIKV infection using a simple statistical model. As reported elsewhere, ZIKV infection was often internationally spread by mildly infected travellers (Tappe et al., 2014; Kutsuna et al., 2014; Korhonen et al., 2016). Potentially high importation risk in many temperate countries has motivated us to explore the risks of importation and local transmission fuelled by travellers. Our model is not as sophisticated as mapping precise risk of transmission using seasonal population dynamics of Aedes species and temperature/climatological data (Bogoch et al., 2016; Nah et al., in press), but the findings at country levels from our study are broadly consistent with what has been briefly described at finer scales (Bogoch et al., 2016). Without using finer scale spatial data (e.g., ecological data on Geographic Information Systems), our approach has crudely and clearly distinguished the risk of local transmission from importation risk at country levels using a more tractable approach. We have shown that the predicted risk of local transmission was frequently seen in tropical and subtropical countries with DENV or CHIKV epidemic experience, while the risk of importation was more scattered around the world. The diagnostic performance of risk model for local transmission was well supported by AUC value of 0.90.

Our study contributes to the risk assessment practice at each country level using estimated risks of importation and local transmission expressed as probabilities. We have shown that the risk of importation may be high in several countries given continued global travel of mildly infected travellers. However, considering that the public health concerns over ZIKV infection stems from the presence of microcephaly, it is more important to focus on the risk of local and widespread transmission that could involve pregnant women. Compared with the risk of importation, the risk of local transmission was particularly highlighted in countries with DENV and CHIKV epidemic experience. Considering that the Olympic Game in Brazil 2016 will elevate the risk of ZIKV to spread over a wider spatial extent (Petersen et al., 2016), the distinction between importation and local transmission will be even more important than it has been recognized. Whereas our risk model of local transmission relied on the presence of transmission by DENV or CHIKV, the use of such a dataset should be deemed only as a proxy. The sensitivity of our best model was 96.4%, and thus there is limited case in which the absence of DENV or CHIKV lead to the local transmission of ZIKV. However, the specificity was limited to 67.7%, indicating that there would be a number of countries in which ZIKV transmission has yet to occur even in the presence of DENV and CHIKV. Our simple model must have missed important additional predictors of local transmission in this regard, and such predictors have to be sought in the future studies. Moreover, to further improve the prediction, ecological data of vector behaviour at finer spatial scale must be incorporated with validation using epidemic data.

A few limitations must be discussed. First, ascertainment bias cannot be ignored for ZIKV infection with substantial fraction of asymptomatic and mild infections. The empirical data in Table 1 must have missed several importation events. Presented risks of importation and local transmission were thus probably underestimated on a whole. Second, the network data we used were static, and the airline transportation data represented only the number of flight routes (i.e., not the number of passengers). Nevertheless, assuming that the geographic patterns of travel have been maintained and only the travel volume has changed over short period of time, our simplistic approach is still justifiable to predict the global spread from 2015. Third, country-specific modelling exercise suffers from heterogeneity within each single country. For instance, the United States, China and Australia have both high and low risk areas of transmission due to vast land with different climatological areas. Finally, we estimated the importation risk of ZIKV in a country as a risk of importing ZIKV from Brazil to the country. Therefore, our model cannot capture the risk of importing ZIKV from other endemic area such as South Pacific.

Despite a clear need to improve predictions in the future, the present study successfully devised a simple global risk prediction of importation and local transmission. Countries with DENV and CHIKV epidemic experience are likely to be at particular high risk, and such countries should be prepared for vector control measures such as avoiding daytime biting and the use of mosquito repellent. To further improve model predictions, it is essential to have laboratory capacity built up in every single country at risk of local transmission.

Conclusions

Risks of importation and local transmission of Zika virus infection were estimated, analyzing epidemiological, ecological and mobility data. Whereas the risk of importation was well explained by the airline transportation network data, the risk of local transmission appeared to be best captured by additionally accounting for the presence of an epidemic of dengue and chikungunya viruses.

Supplemental Information

Supplemental Information 1 Data source of the data of Zika virus importation

Click here for additional data file.

Supplemental Information 2 Effective distance from Brazil

Country-specific effective distance from Brazil is shown.

Click here for additional data file.

Additional Information and Declarations

Competing Interests

Author Contributions

Data Availability

The authors declare there are no competing interests.

Kyeongah Nah performed the experiments, analyzed the data, contributed reagents/materials/analysis tools, wrote the paper, prepared figures and/or tables, reviewed drafts of the paper.

Kenji Mizumoto analyzed the data, contributed reagents/materials/analysis tools, wrote the paper, prepared figures and/or tables.

Yuichiro Miyamatsu performed the experiments, analyzed the data, contributed reagents/materials/analysis tools, wrote the paper, prepared figures and/or tables.

Yohei Yasuda and Ryo Kinoshita analyzed the data, wrote the paper, prepared figures and/or tables.

Hiroshi Nishiura conceived and designed the experiments, performed the experiments, analyzed the data, wrote the paper, reviewed drafts of the paper.

The following information was supplied regarding data availability:

The raw data were supplied as Supplemental Information.

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
