# Peer review of "Estimating risks of importation and local transmission of Zika virus infection"

_PeerJ, doi:10.7717/peerj.1904_

## Round 0.1 · original submission · Major Revisions

Both reviewers suggested a number of revisions to improve the manuscript. Please address their comments carefully and submit your revised study.

Kind regards,

·

Basic reporting

Nah and colleagues use a statistical model based on survival analysis to estimate the worldwide risk of importation of Zika virus to countries not yet affected. To assess the reliability of their importation risk estimates, authors use a standard diagnostic tool (ROC curve).

The introduction is relatively clear, with the following suggestions:
- dates should be mentioned when referring to the current Zika outbreak (line 37-38)
- the link between microcephaly and Zika is still controversial: this controversy could be mentioned with appropriate references (ref 1 is not relevant in my opinion)

In the abstract, I would avoid using “area under the curve”, which is probably not clear for readers not versed in survival analysis.

The raw data are not supplied but they should be: data essentially consist of the airline routes publically available, so there are no confidentiality/proprietary issues. The data used for this study, as well as the computer code, should be made available such that replication attempts are facilitated.

Experimental design

The research question is well defined and is the first attempt (to my knowledge) to estimate Zika virus importation risk with a relatively simple statistical model, which is of great interest. The use of survival analysis to investigate the risk of importation is, a priori, appropriate.

As a general comment, the Methods section should be described more clearly and with more details. My specific comments are:
- “by dimension of 3 times 4,600…” lines 63-64: this is not clear, please explain. I also suggest you use “airports” and “flight routes” instead of “nodes” and “edges”.

- Mosquitos abundance “was analysed” (line 66): how was it analyzed? Or were density values simply retrieved? Please give a brief description of what was actually done.

- Country specific data on CHIKV was retrieved. Why not Dengue (as mentioned in intro)? Ref [3] provides a light narrative description of the similarity between CHIKV and Zika, but one could believe Zika may have been in the footstep of DENV too. Authors may want to either include DENV or justify more clearly why CHIKV was used but not DENV.

- The definition of “P_ij” (line 80) as the fraction of passenger flux is misleading: in the discussion, one of the limitations stated is that airline data “represented only the number of flight routes (i.e. not the number of passengers)” (line 177). Please clarify.

- Variable “m_j” is not defined.

- Function “S_j” is not defined.

- The methods describing what was done for the risk of local transmission are not clear enough in my opinion. Please expand.

- What is the horizon of the risk of importation?

Validity of the findings

Given the data and computer code used are not provided and the gaps in the methods descriptions (see above), the actual results presented in this manuscript could not be fully reviewed. However, the results presented seem plausible.

The main comment I have is on the way the results are presented. Using Uganda 1947 as the temporal origin should be at least relegated in supplement material. Only using Brazil 2015 as origin makes sense for me. Indeed, airline data as of 2014 are used to define the hazard function. How can this data be relevant for extrapolating the hazard prevailing from 1947 to the 1980s? Hence, I suggest not presenting results using Uganda as the origin. This will make the results of this manuscript clearer to read and interpret.

In the discussion, there are two sentences that should be tamed:

- line 159: “unavoidable” sounds too strong. Maybe a table of probabilities of importation risk at various times in the future for representative countries would help the readers assessing if it’s really unavoidable.

- line 186-187: given the link between microcephaly and Zika infection is still controversial, authors may want to avoid asserting it implicitly.

Finally, comparing the present findings with previously published studies estimating similar Zika virus importation risks (for example Bogoch, Lancet 2016) would help readers to form a global opinion about such estimations.

Additional comments

The research question is clearly of interest, especially now that we are still in the “fog of war” regarding Zika virus epidemiology.

The methods are not presented adequately and the paper is not reproducible in its current state.

Findings are interesting, but the results using Uganda as time origin should not be in the main text, in my opinion.

·

Basic reporting

See "General Comments for the Author"

Experimental design

See "General Comments for the Author"

Validity of the findings

See "General Comments for the Author"

Additional comments

General comments
The manuscript is an interesting short communication about the risk of importation and local transmission of ZIKV. The probability of introduction was estimated using the inverse of the effective distance from ZIKV supposed “origin” location. The local transmission was predicted using the abundance (?) of Aedes mosquitoes or the presence(?) of CHIKV local transmission. These risk probability estimates can be used by government agency to communicate the ZIKV risk to the population. However, some part of the methodology is explained in an suboptimal way. Furthermore the language is often clumsy. Simplicity in the approach does not mean that the methodology has to be described with a sloppy attitude and language. The authors need to improve the manuscript before it can be considered for publication.

Main points:
It seems to me that the “effective distance” is defined in two not-compatible ways. dij is 1- the logarithm of the fraction of passenger flux reaching locality i from locality j, while mji is defined as the minimum sum of effective lengths along the legs of the path. Aren't they to different metrics describing the effective distance? How they can be merged in the same predictor?
Another potential problem that I see is that the author decided to change the “root” point of ZIKV infection from Uganda 1946 to Brazil 2015. In this way the authors excluded from the analysis a big set of countries with a high risk of importation and maybe also risk of local transmission. These countries are coloured in grey in Figure 2A and B. The authors should address this weakness in the text.
To estimate local transmission the authors multiplied the probability of a country to import ZIKV for the presence of Aedes species or CHIKV (as written in line 104:108). These variables are described as binomial, presence or absence, therefore what's the point in the multiplication?
Moreover, in line 106 the authors equal the observed presence of local transmission with the dichotomous variable, that is instead the presence of CHIKV or Aedes species. Please explain better what you mean.
In row 170 you say that “The absence of CHIKV transmission does not necessarily indicate that ZIKV cannot cause local transmission, and in fact, the specificity value of CHIKV-based risk of local transmission was 81.5%.” I do not understand how a low specificity of the CHIKV-based model indicates that ZIKV local transmission is possible even though CHIKV is not there. Specificity is the proportion of negatives that are identified as such. 81% specificity means that your CHIKV- based model predicts some location as positive for ZIKV when they are negative. Therefore, at most you can derive that the presence of CHIKV local transmission does not mean that ZIKV is also transmitted. This is exactly the opposite of what you stated. Explain better what you mean.
In figure 2B, you write “local transmission risk of countries affected by Chickungunya” but again you should stress that you excluded all the countries that had ZIKV before the outbreak in Brazil.

Minor points:
Abstract: in general it needs a language check. For example “An epidemiological dataset of country-specific importation events”. Or “Aedes species in common”. The discussion part is poor and uninteresting.
Row 25 to 28: Not clear what you mean with this sentence? Who characterized the risk of importation? Do you mean explained? Rephrase
Row 32: Non sense sentence: “In the presence of sustained local transmission”
Row 40: not only sharing… Rephrase, bad English
Row 43: What do you mean: “with abundance of Aedes species” Probably you mean with abundant population of Aedes mosquitoes. Try to use the right terminology, otherwise the language sounds sloppy.
Row 51: Publicly not publically.
Row 54 mosquitoes not mosquitos
Row: Screening countries? I'll stop here to correct the English.
Row 64: Not clear what you mean with “Dimension of 3 times”
Row 59: the mid-point of available time window
Row 65: Aedes Africana, the species name is always lower-case
Row 67: “local transmission of ZIKV infection occurs only where the vector species are abundant”. Please justify this statement. The probability of transmission for vector borne disease agents is usually proportional to the rate of infected mosquito biting human being. The abundance of the vector is only one of the interplaying factors.
Row 94: What do you mean with “censored observation”?
Results: report the ML estimate of k.
Row 103: Please add a reference number to the formula. Furthermore, is the time multiplying the inverse of the effective distance or it is a subscript of k parameter?
Row 106: It is not clear to me what data you used to assess the risk of local transmission. Did you use another database or data source reporting the local transmission of ZIKV?
Row 111: Use capital letter when indicating acronyms.
121 to 124: Rephrase this sentence, it is confusing.
137: Rephrase “Countries in the absence...”
140: the top
146: risk not risks

---

## Round 0.2 · Minor Revisions

Dear Authors,

Both reviewers appreciated the quality of your revised manuscript. The paper is now suitable for publication on PeerJ pending remaining minor revisions.

Concerning Introduction, please highlight the crucial issue of Aedes control programs for prevention of Zika outbreaks, you may consider to cite this recent review, which contains useful details: http://link.springer.com/article/10.1007%2Fs00436-016-4971-z.

Kind regards,

·

Basic reporting

This second version is significantly improved in terms of clarity of the methods and readability of the results. Numerical values of the results were slightly changed because of the changes in the methods (e.g. use of presence of dengue virus), but the overall message of this study is still the same. This study is reproducible now (all data are accessible, methods described with enough details), although data given as supplemental files could be improved (that is, make it easier for someone wanting to replicate this work)

I have no major issues and some minor ones are listed hereafter.

Experimental design

No comments

Validity of the findings

No comments

Additional comments

- Line 99: define n_i (airport #i ?)
- Equation (1): “L=”, not “L–”. Also, although it’s obvious given the context, authors may want to change the notation of “L” which represents two different quantities (path length (1) and also likelihood (6))
- Equation (7): p_j instead of p
- Line 175: “were” not “was”

·

Basic reporting

The authors reviewed the manuscript following the reviewers suggestions. However there are incongruences and mismatches in the results that have to be fixed before publication.

Experimental design

Equation 5 is not correctly visualized ( I see a series of question marks instead of the formula).

It's not clear what mji is in the formula in line 102. It may be confusing to use m to indicate the number of flights and the effective distance at the same time.

Validity of the findings

You should add in the limitation paragraph that your survival analysis approach does not take in account the fact that the virus can be imported also from countries that have received it from Brazil. Therefore, it maybe that the estimate of the probability of importation has been underestimate for some countries (as well as the estimated local transmission risk).

Furthermore, even after considering all the limitation listed in the discussion, I am not able to interpret some of the results, such as the very high risk of importation and local transmission in Czech Republic (Figure 2AB). I do not think that this country has many (direct) flights from Brazil (maybe I'm wrong but I'd be surprised), while for the risk of local transmission, the only Aedes species there is albopictus that has been introduced but is not established. There are no CHIKV reported cases while Dengue has only been imported but not locally transmitted.

Furthermore, there are mismatches between Figure 2 and 3. For example in Figure 2 France is reported with a lower risk of transmission than Czech Republic while in Figure 3 France is reported among the 30 countries with the highest estimated risk of local transmission but Czech Republic is not.

Please fix these (and other potential) mismatches.

Additional comments

The authors reviewed the manuscript following the reviewers suggestions. However there are incongruences and mismatches in the results that have to be fixed before publication.

---

## Round 0.3 · Minor Revisions

Dear Authors,

I am happy to inform you that both Reviewers appreciated your revision and now supported publication of your manuscript.

From my own reading, I feel that the Introduction is too brief and the Zika virus background needs to be improved a bit. Please revise and consider giving some emphasis to the importance of the control pf Aedes vectors for Zika outbreak prevention.

Kind regards,

·

Basic reporting

No comments

Experimental design

No comments

Validity of the findings

No comments

Additional comments

No comments

·

Basic reporting

Manuscript is ready for publication.

Experimental design

Manuscript ready for publication.

Validity of the findings

Manuscript ready for publication.

Additional comments

Manuscript ready for publication.

---

## Round 0.4 · accepted · Accept

The manuscript can be now accepted as it.